Wildlife overpass structure size, distribution, effectiveness, and adherence to expert design recommendations

Brennan Liam 1 2
Chow Emily 1
http://orcid.org/0000-0002-1961-0509 Lamb Clayton 3 ctlamb@ualberta.ca
1 British Columbia Ministry of Forests, Lands, and Natural Resource Operations and Rural Development , Cranbrook, British Columbia , Canada
2 The University of British Columbia , Vancouver, British Columbia , Canada
3 The University of British Columbia , Kelowna, British Columbia , Canada
Pimm Stuart
Electronic publication date: 2022 Dec 12
Publication date: 2022
Volume: 10
Electronic Location ID: e14371
Received 2022 Jul 13; Accepted 2022 Oct 19
Copyright: © 2022 Brennan et al.
Copyright year: 2022
Copyright holder: Brennan et al.
License: This is an open access article distributed under the terms of the Creative Commons Attribution License, which permits unrestricted use, distribution, reproduction and adaptation in any medium and for any purpose provided that it is properly attributed. For attribution, the original author(s), title, publication source (PeerJ) and either DOI or URL of the article must be cited.
License URL: https://creativecommons.org/licenses/by/4.0/

Keywords: Highway, Wildlife, Crossing structure, Overpass, Effectiveness, Guidelines

Funding: Liber Ero Fellowship Clayton Lamb was supported by the Liber Ero Fellowship while conducting this work. The funders had no role in study design, data collection and analysis, decision to publish, or preparation of the manuscript.

==============================
It is now well evidenced that wildlife crossing structures paired with exclusion fencing reduce wildlife vehicles collisions while facilitating wildlife connectivity across roadways. Managing animal mortality and subpopulation connectivity is crucial to successful species and landscape stewardship. Highway mitigation projects are large economic investments that remain on the landscape for many decades. Governments and planning agents thus strive to balance cost and benefit to build cost-effective structures with the greatest positive impact on local wildlife and motorists. Ideal dimensions of overpasses and underpasses vary by species, but scientists generally suggest that overpasses for large mammals should be approximately 50 m wide. Optimal width also depends on structure length, with longer structures requiring additional width. Accordingly, experts have suggested a width to length ratio of 0.8. We sought to assess how these recommendations manifested in practice—where agencies use this information to design and build structures while also balancing cost and logistical challenges—and the degree to which built structures conform to current recommendations. We identified 120 wildlife overpasses across the world that were constructed to reduce the negative impacts of roads. Using a novel measurement technique, we analyzed the dimensions of these 120 overpasses located in North America, Europe, Asia, and Oceania. The average width of the wildlife overpasses was 34 m. Most wildlife overpasses located in North America and Europe did not meet their respective dimensional expert guidelines. We investigate reasons explaining the non-compliance and provide recommendations for future overpass designs. Building on previous evidence showing that wider overpasses have increased crossing rates, we examined crossing rates for multiple large mammal species across 12 overpasses located in western North America. We qualitatively observed that wider North American overpasses (40–60 m), in or near compliance with expert guidelines, were associated with a more diverse set of species use and had nearly twice the average crossing rates when compared to non-compliant, narrow North American overpasses. In reviewing various studies from around the world, we conclude that wide overpasses (~50 m) continue to present ecologically sound and cost-effective solutions for decreasing the barrier of roadways, especially when targeting width-sensitive species and large assemblages of mammals. Future studies, however, are encouraged to further explore the specific instances when underpasses and narrower overpasses present more cost-effective ecological solutions, or how these structures can complement wide overpasses in successful wildlife crossing systems.

Introduction

Background

Roads provide essential connection corridors for people and goods across the world but can be challenging features for wildlife to cross safely and have myriad environmental consequences (Forman & Alexander, 1998). Decades of research has shown that roads can degrade and fragment habitat, create barriers to animal movement, and be a major source of animal mortality (reviewed by Fahrig & Rytwinski, 2009; Trombulak & Frissell, 2000). With naturally low population densities and large home ranges, large mammal populations are especially vulnerable to the potentially exacerbating threats of wildlife vehicle collisions (WVCs) resulting in mortality (Ceia-Hasse et al., 2017; Gunson, Clevenger & Hall, 2003; Kusak, Huber & Frkovic, 2000) and the genetic isolation of sub-populations on either side of the road (Riley et al., 2006; Proctor et al., 2012; Sawaya, Clevenger & Schwartz, 2019). Creative solutions to combat these threats are needed, especially on highways where traffic volumes are high and collisions are especially dangerous for wildlife and people.

A solution to WVC’s that is gaining in popularity is the construction of wildlife crossing structures (Sijtsma et al., 2020). When paired with wildlife exclusion fencing, wildlife crossing structures reduce wildlife mortality from WVCs (reviewed by Huijser et al., 2009) while still promoting demographic and genetic connectivity for wildlife across the road (Sawaya, Clevenger & Kalinowski, 2013; Sawaya, Kalinowski & Clevenger, 2014). These structures are often met with little to no public resistance because they provide added benefits to motorists (e.g., increased safety and reduced costs related to WVCs) without altering the flow of traffic (reviewed by Huijser et al., 2009; Sijtsma et al., 2020). When wildlife crossing structures are paired with adequate wildlife fencing, studies have found an approximate 86% decrease in reported WVCs (Huijser et al., 2009). For example, in Banff National Park, a series of crossing structures and fencing along a 23 km section of the Trans-Canada Highway reduced wildlife collisions by 80%, and reduced collisions with common species such as deer and elk by 96% (Clevenger, Chruszcz & Gunson, 2001).

Overpass vs underpass

Wildlife crossing structures can be subdivided into two classes: underpasses and overpasses, which respectively allow wildlife to cross under or over the roadway (Clevenger & Huijser, 2011). Clevenger & Huijser (2011) suggest that large wildlife overpasses are an optimal crossing structure choice for a diversity of species. Furthermore, various studies have shown that carnivores and ungulates (e.g., grizzly bears, moose, wolves, deer, elk, pronghorn, and desert bighorn sheep) prefer large, open overpasses compared to more constricted underpasses (Clevenger & Waltho, 2005; Kusak et al., 2009; Sawyer, Rodgers & Hart, 2016). Ultimately, however, various abiotic and biotic factors influence the use and effectiveness of a crossing structure (Clevenger & Waltho, 2000; Clevenger & Waltho, 2005; Barrueto, Ford & Clevenger, 2014; Seo et al., 2021). Consequently, a universal species-specific preference for overpasses vs underpasses can be difficult if not impossible to clearly establish. For example, Simpson et al. (2016) and Sawyer, Rodgers & Hart (2016) present contradictory mule deer preference for overpasses vs underpasses in Nevada and Wyoming, U.S.A., respectively. Similarly, while many large mammals in Banff National Park exhibited a preference for overpasses, Gloyne & Clevenger (2001), found a clear cougar preference for underpasses. Based in part on various species-specific crossing structure preferences, many highway wildlife crossing structure projects employ both structure types along the length of a project, promoting permeability for the greatest number of species (Clevenger & Waltho, 2000; Clevenger & Waltho, 2005; Cramer, 2012).

Overpass width

Among many considerations when designing a crossing structure (e.g., location and adjacent land use), one important aspect is the structural dimensions of the structure. For the purposes of this study, we define wildlife overpasses as above-grade structures that cross over roads and/or other transportation infrastructure. Similar terms used in literature to describe overpasses include green bridges and eco-ducts. In comparison to wildlife underpasses, overpasses are preferred by most large mammals (Kusak et al., 2009; Sawyer, Rodgers & Hart, 2016), but often also cost significantly more (McGuire & Morrall, 2000).

Overpass width is an important design consideration. Studies that have compared overpass effectiveness data with overpass width suggest that wider overpasses more effectively enable the crossing of large mammals opposed to narrow structures that may deter crossings if animals feel uncomfortable and hesitant to cross (Pfister et al., 1997; Clevenger & Waltho, 2005). However, wildlife overpasses are costly structures, often 5–15 million dollars USD and wider structures cost more than narrower ones. Transportation agencies thus attempt to build structures whose dimensions satisfy the ecological role for which the structures are designed to support while delivering the project in a cost-effective manner. While the ideal width of a wildlife overpass will depend upon various factors (e.g., project objectives, target species, location etc.), in North America the recommended highway overpass structures width is 50–70 m over four lane highways (Clevenger & Huijser, 2011). Overpass widths should also consider the length of the structure, where longer structures require additional width. This consideration is addressed by the recommendation of a width to length ratio >0.8 for overpasses in Europe (Iuell et al., 2003).

These recommendations are supported by research that suggests that some large mammals prefer to cross an overpass that is at least 50 m wide. Using wildlife crossings rates and demographic analysis, Ford, Barrueto & Clevenger (2017) suggest that in comparison to underpasses, large overpasses (>50 m in width) serve as crucial passages for family units of grizzly bears whose survival is crucial to population viability. Also, in Banff National Park, the crossing rates of various large mammals were positively linked to structure width, and negatively associated with structure length (Clevenger & Waltho, 2005). Similar evidence from (Pfister et al., 1997) shows that crossing structures in Europe that were less than 20 m were used significantly less than wider structures, and that between 20 and 50 m the crossing rate of wildlife increased and subsequently flattened out.

In practice, countless factors (e.g., political interest, budget, government structure, etc.) other than expert recommendations ultimately influence the width of overpass structures (Woo et al., 2018). In South Korea for example, researchers found that some of the technical parameters of crossing structures (e.g., underpass height, underpass openness ratio) often fail to comply with expert recommendations (Woo et al., 2018). To further investigate overpass dimensions and their adherence to expert guidelines, our study compiled a global inventory of wildlife overpasses and compared their physical characteristics to various expert recommendations/guidelines.

Crossing structure effectiveness

To properly assess the effectiveness of a wildlife crossing structure, the ecological role of wildlife structures must first be defined (Clevenger, 2005). While ambitious, the goal of highway mitigation programs should be to decrease the negative ecological effects of the roadway and improve wildlife population viability (van der Grift et al., 2013). However, studies that directly demonstrate wildlife crossing structures increase the population viability of wildlife are scarce and have not been performed for large terrestrial mammals (van der Ree et al., 2007; van der Ree et al., 2009). Changes in population viability are difficult to measure/model (van der Grift et al., 2013); so many studies instead rely upon two related metrics of wildlife crossings structure success: namely, the surrounding reduction of wildlife mortality as a result of fencing that excludes wildlife from collisions on the roadway (Huijser et al., 2009, 2016; Rytwinski et al., 2016) and, the ability of the structure to promote roadway permeability and dispersal between subpopulations on either side of the road (i.e., successful crossings, Kusak et al., 2009; McKinney & Smith, 2007; Sawaya, Clevenger & Kalinowski, 2013; Sawaya, Kalinowski & Clevenger, 2014).

For the purposes of this study, we concentrate on the ability of the wildlife overpasses to reduce the barrier effect of the road through movement over the roadway. While other studies have measured overpass effectiveness in terms of demographic and genetic connectivity (Sawaya, Clevenger & Kalinowski, 2013; Sawaya, Kalinowski & Clevenger, 2014; Ford, Barrueto & Clevenger, 2017) or percent of successful crossings (Denneboom, Bar-Massada & Shwartz, 2021), our study amasses wildlife crossing rates for wildlife overpasses in western North America. Wildlife crossing rates are the most commonly used wildlife overpass effectiveness metric (van der Grift et al., 2013) and provide evidence of roadway permeability and overpass use, although can be difficult to compare between areas with differing abundance of wildlife. Consequently, we sought to answer the following questions (1) can we gather sufficient crossing data from state and provincial agencies to compare crossing rates in relation to overpass dimensions? and, (2) building off the results from this review and lessons learned in analyzing the crossing data, can we recommend sampling designs that could help measure structure effectiveness in the future?

Road agencies are increasingly in need of evidence to support the design and construction of wildlife overpasses. This study was designed to help inform road ecologists and transportation agencies as they design wildlife overpasses. First, we compiled a global inventory of wildlife overpasses to understand the width of overpasses built to date, and to what extent these widths adhere to expert guidelines. Next, we performed an analysis of overpass effectiveness in relation to width for various structures in northwestern North America. In summarizing overpass widths, guideline compliance, and performance, we present the current evidence for the optimal width of wildlife overpasses and suggest a robust monitoring approach to assess their effectiveness in the future.

Materials and Methods

Literature review and identifying overpasses

A literature review was conducted to locate and identify overpasses worldwide and measure their structural dimensions. Keywords included “wildlife overpass”, “eco-duct”, “green bridge”, and “wildlife crossing structure”. The search engines Google and Google Scholar were used to collect pertinent peer-reviewed literature, government reports, and websites, that contained relevant information about overpasses around the globe. Further consultation with various experts in the field supplemented the inventory with additional structures. Subsequently, location data collected in the literature review was used to locate the structures in Google Earth Pro 7.3.4.8573 (64-bit). Overpasses were included in the review if they were visible on Google Earth Pro, if sufficient supporting information was available to locate the structures, and if the apparent primary purpose of the structure was for use by wildlife. Multi-use overpasses, as described in Clevenger & Huijser (2011), (e.g., overpasses visibly paved with asphalt), which target use by humans and wildlife, were not included in the inventory. Similarly, because landscape bridges (overpass structures >80 m wide) aim to restore habitat, biodiversity, and connectivity at a larger scale, they have different ecological objectives than wildlife overpasses and were also excluded from the review (Iuell et al., 2003). To explore the relationship between overpass dimensions and wildlife crossing rates, species-specific wildlife crossing data was also collected from literature and government reports for a sample of 12 overpasses located in western North America. Appropriate transportation professionals were contacted to obtain overpass dimensions and crossing data when not readily available in the literature.

Measuring dimensions

Overpasses can vary greatly in size, shape and design (Iuell et al., 2003; Sołowczuk, 2020). For the purposes of accurate comparison between different structures, we developed overpass dimensional definitions for all measurements made in Google Earth Pro. If the aerial image resolution was sufficient and the view was unobstructed, the estimated width(s) and length(s) were measured in Google Earth Pro. Reported dimensions were also recorded from available literature or government reports. Government agencies and reports often only reported a width and length and often lacked supporting information that explained how the structural dimensions were obtained. While the width of the overpass is often clearly visible from aerial imagery, the length can be difficult to distinguish especially with older overpasses where plant communities have extensively re-colonized the soil on and around the structure. In Google Earth Pro, we estimated three overpass length measurements: the width of roadway below (including median and lanes of traffic), the length of the overpass headwall, and the length of the overpass including visible ramps. We recorded two width measurements: the inner edge to edge width between opposite headwalls or fencing, and the outer edge to edge width between opposite headwalls. All measurements are clearly illustrated in Fig. 1. In accordance with recommendations made by Iuell et al. (2003), we define our primary overpass width as the inner width of wildlife overpass structures. Likewise, we define the overpass length as the headwall length because it is a consistent, representative length metric amongst the majority of overpasses, and is easily viewed in aerial imagery.

Figure 1 Graphical representation of the measurement procedure for all overpass dimensions obtained in Google Earth Pro 7.3.4.8573 (64-bit).

(A) Inner width, (B) outer width, (C) total length (including landscaped ramps), (D) headwall length, (E) roadway width.

Harrington et al. (2017) compared Google Earth Pro path measurements to physical measurements of road features and found evidence to support the use of Google Earth Pro as a scientific measurement tool. We also tested the accuracy of the ruler tool on objects of known length, 91.44 m (100 yard) long football fields, to assess reliability. Across 20 NFL and NCAA football fields, we found an average error rate of 0.2% (see Supplementary Information, Table S8). Because elevation change along structures is minimal it was assumed to have a negligible effect on measurement.

Overpass crossing rates

We assessed the wildlife crossings effectiveness of twelve overpasses located within a 500 km radius circle in northwestern North America, inclusive of British Columbia (Canada) Alberta (Canada), Montana (U.S.A.), and Washington (U.S.A) (Fig. 2). The 12 overpasses were selected for our analysis because they are used by similar assemblages of large mammals in adjacent, interconnected montane ecoregions. The availability of crossing rate data, and variety of overpass widths included in the sample was favourable to investigate a possible relationship between overpass width and wildlife crossing rates. Using multiple camera traps (minimum two) each monitoring project recorded the number of successful passages across the overpass for a variety of ungulate and carnivore species (minimum of 164 monitoring days and maximum of 3,180 monitoring days). We gathered data for ten large mammal species commonly found in the montane ecoregion of western North America: black bears (Ursus americanus), grizzly bears (Ursus arctos), wolves (Canis lupus), coyote (Canis latrans), cougars (Puma concolor), deer (Odocoileus sp.), elk (Cervus elaphus), moose (Alces alces) and, bighorn sheep (Ovis canadensis). Like Ford, Barrueto & Clevenger (2017), we evaluated the species-specific number of successful crossings per monitoring day to develop a metric that could be compared between structures. Subsequently, a taxa-specific, and total crossings linear regression was performed for all twelve structures in relation to their inner width and inner width: headwall length ratio as estimated with Google Earth Pro. We did not have information on relative abundance of each species around each structure which we acknowledge limits the inferential power of this approach.

Figure 2 Location in Western North America of the 12 wildlife overpasses included in overpass effectiveness analysis.

500 km radius shown in dashed line. Base maps source: http://maps.stamen.com, CC-BY 3.0.

Results

We identified 120 individual wildlife overpasses across the world (Fig. 3). The majority of the 120 wildlife overpasses were concentrated in northern latitudes (n = 73) (Latitude >40°) and G20 Nations such as Canada (n = 12), the United States (n = 16), Germany (n = 11), Holland (n = 24) and South Korea (n = 35).

Figure 3 (A) Map of the 120 wildlife overpasses found in the literature review.

Clusters of overpasses were found in (B) western North America, and (C) Europe and eastern Asia. Base maps source: http://maps.stamen.com, CC-BY 3.0.

While a date of construction was not recorded for all structures, the structures for which a construction date was available in the literature were built between 1975–2019. Across the 120 overpasses, the average width was 34 m, an average of four traffic lanes were crossed, and an average width to length ratio of 0.58 was observed (Table 1). In North America, only 29% of overpass structures adhered to the >50 m width recommendation. Similarly, in Europe, only 50% and 21% of structures were respectively in compliance with the >50 m width, and >0.8 width: length ratio recommendations. Fig. 4 shows the distribution of overpass widths, highlighting a slight skew towards larger widths.

Table 1 Mean overpass parameters measured in Google Earth Pro 7.3.4.8573 (64-bit), associated expert recommendations, and compliance rate of (a) Global, (b) North American and (c) European structures.

	Wildlife overpass parameters (m)	Expert recommendations3,4 (m)	Compliance rate (%)	
a: Global	
Mean width (n = 97)1	34 (3–76)	-	-	
Mean length (n = 90)2	65 (21–138)	-	-	
Mean width: length ratio (n = 90)	0.58 (0.06–2.76)*	-	-	
b: North America	
Mean width (n = 28)1	33 (6–65)	>50	29	
Mean length (n = 27)2	62 (29–109)	-	-	
Mean width: length ratio (n = 27)	0.53 (0.09–1.10)*	-	-	
c: Europe	
Mean width (n = 52)1	38 (11–76)	>40	50	
Mean length (n = 52)2	72 (23–138)	-	-	
Mean width: length ratio (n = 52)	0.60 (0.13–2.76)*	>0.8*	21	
Notes:

1 Estimated inner width of overpass using ruler tool in Google Earth Pro 7.3.4.8573 (64-bit).

2 Estimated headwall length of overpass structures using ruler tool in Google Earth Pro 7.3.4.8573 (64-bit).

3 Expert width recommendation of 50 m or greater for overpasses in North American (Clevenger & Huijser, 2011), similar recommendations in Iuell et al. (2003).

4 Expert width: length recommendations of 0.8 or greater for overpasses in Europe (Iuell et al., 2003).

* Unitless ratio.

Figure 4 Dimensions of 120 wildlife overpasses from around the world included in the review.

All overpass measurements performed using the ruler tool in Google Earth Pro 7.3.4.8573 (64-bit). We define width as the estimated inner width of overpass and length as the estimated headwall length of overpass structures.

For the sample of twelve northwestern North American overpasses, the average crossing rate for 40–60 m wide structures was 1.6 (SE = 0.4) animals per day while structures <40 m was 0.7 (SE = 0.4) animals per day. The total number of large mammal crossings per day was positively related (β = 0.013, SE = 0.013) to width, but the effect was not significant (p = 0.35), likely due to the small sample sizes available. Similarly, the total number of large mammal crossings per day was positively related (β = 1.16, SE = 0.97) to width: length ratio and the effect was not significant (p = 0.26) (Fig. 5). A non-significant positive relationship (p = 0.26) between taxa-specific crossing rates were also found in relation to overpass width and width:length ratios. Among the nine species included in our review, we found a species-specific response to overpass width (see Figs. S1 and S2). Of the large mammals (n = 9) included in the study, we found that the larger overpasses (n = 8) between 40–60 m wide crossed an average of 6.8 species in comparison to narrower structures (n = 4) less than 10 m wide that crossed an average of three different species.

Figure 5 Overpass inner width, and W:L ratio in relation to the number of successful wildlife crossings per month. Data compiled from transportation agencies and government reports.

Each dot represents an overpass structure. Species included in analysis: (black bears (Ursus americanus), grizzly bears (Ursus arctos),wolves (Canis lupus), coyote (Canis latrans), cougars (Puma concolor), deer (Odocoileus sp.), elk (Cervus elaphus) moose (Alces alces) and bighorn sheep (Ovis canadensis) crossing rates and the width, W:L ration of 12 overpasses located in western North America. See Table S7 for details.

Discussion

Distribution of wildlife overpasses

Over the past 40 years, many wildlife overpasses have been constructed across the world with the objective of increasing the safety of local wildlife species, motorists, and restoring habitat connectivity. In total, we identified 120 structures located in North America, Europe, southeast Asia, South America and Australia. Where habitat fragmentation has caused significant wildlife and biodiversity declines, (e.g., Western Europe and parts of Asia) governments have responded by investing in landscape connectivity programs that include crossing structures (Sijtsma et al., 2020; Woo et al., 2018). We found especially high concentrations of wildlife overpasses in areas of high road and human density such as South Korea and Holland. Many overpasses were also identified in western North America that were built to support the local assemblage of large bodied mammals such as grizzly bear, moose, and elk. We found few records of overpasses in many of the worlds’ most abundant areas of large mammals such as southern Africa and South America, although a single structure was located in Argentina.

Overpass widths

Of the overpasses for which Pimm et al. (2021) were able to find supporting dimensions information (n = 82), they found that nearly half of the overpasses were greater than 50 m wide. In contrast, we found only 20% of the overpasses included in our review (n = 97), were wider than 50 m. We suspect the discrepancy of results is due to two main factors. To allow for comparison to relevant guidelines we excluded any overpasses that were >80 m wide (classified as a landscape bridge (Iuell et al., 2003), reducing the percent of the overpass structures that are >50 m wide. Had landscape bridges been included in our review, however, only an additional 6% of structures would have been greater than 50 m wide. Secondly, we found that Google Earth Pro measurements of the inner width of the structures for which we had sufficient supporting information (n = 24) were 7% less than those reported in the literature. With very few mentions of overpass dimensional definitions in the literature, we suspect transportation professionals and experts often use what we have defined as the outer width of the structure, possibly explaining the Pimm et al. (2021) results. Moving forward, a clear definition of overpass width is needed, especially where wildlife overpasses incorporate elements such as fencing and earthen berms to limit acoustic impacts of the road (Sołowczuk, 2020). As suggested here, and by Iuell et al. (2003) the inner width of the overpass is most representative of the width available to animals as they cross the structure and should be the default width measurement used by transportation agencies. Universal definitions of both overpass width and length will help to ensure that future wildlife overpasses meet expert recommendations and fulfill the ecological role for which they are designed.

Adherence to expert-based recommendations

Comparing the built dimensions of overpasses to expert recommendations, we found that wildlife overpasses were generally built narrower than standards recommended by experts. The goal of dimensional standards, outlined by Clevenger & Huijser (2011) and Iuell et al. (2003) are to facilitate overpass designs that are effective in achieving sufficient crossings of target species and to provide adequate conservation return on infrastructure investments. Notably, 71% of North American overpasses (n = 28) were in non-compliance with the >50 m wide recommendation (Clevenger & Huijser, 2011). Clevenger & Huijser (2011), however, did not explicitly define how widths should be measured. As mentioned above, we suspect many transportation experts and professionals have historically relied on the outer width of the overpass structure as opposed to what we define as the overpass inner width. However, even when comparing the >50 m recommendation to the outer width of the 28 North American structures, a high non-compliance of 64% was still observed. Similarly, 50% of European overpasses failed to meet the respective >40 m inner width recommendation (Iuell et al., 2003). In stark contrast, compared to their respective expert guidelines, Woo et al. (2018) found a 14% rate of non-compliance of overpass widths in South Korea. In both North America and Europe, the non-compliance of overpass width may partially be explained by the specific targeting of certain ungulate species such as deer which are less sensitive to more narrow overpasses (Iuell et al., 2003). It should also be acknowledged that many structures included were the first of their kind and were constructed before many of the science-based recommendations were available.

Notably, 79% of the European structures in this review did not meet the recommended minimum 0.8 width: length ratio established by the European Transportation Agency (Iuell et al., 2003). The width to length ratio recommendation is often less prominent, or absent, in both the peer reviewed literature and transportation handbooks on overpass dimension recommendations. As such, the focus on width as a static quantity, rather than a dimension that needs to be considered in concert with length, may be poorly communicated to transportation professionals in the current literature. Future projects should consider that longer overpasses must also be wider to facilitate animal passage.

Effectiveness

The overpass width effectiveness analysis produced a positive but insignificant relationship between overpass width, width: length, and the total number of large bodied mammals, ungulates, carnivore or species-specific crossings per day. This was not altogether surprising given the limited sample size (n = 12), and our inability to control for local wildlife densities. Nevertheless, it was qualitatively observed that wider North American overpasses (40–60 m), in or near compliance with expert guidelines, were associated with a more diverse set of species use and had nearly twice the average crossing rates when compared to non-compliant, narrow North American overpasses. These findings support evidence that suggests wider crossings structures favour the passage of a wider array of taxa (Clevenger & Huijser, 2011), and also support the recommendations of Rytwinski et al. (2015) that more rigorous experimental study designs to monitor the effectiveness of structures would help guide future investments in wildlife crossing infrastructure. Although, we also acknowledge that areas of expected higher ecological value may receive more investment (i.e., wider overpasses) and thus sampling designs that control for local densities and species assemblages around overpasses will be important to implement.

A species-specific response to crossing structure parameters such as width is well established (Mata et al., 2008; Sawyer, Rodgers & Hart, 2016). Researchers have shown conflicting crossing structures width responses in ungulate species such as big-horned sheep, deer, elk and pronghorn (Clevenger & Waltho, 2005; Gagnon et al., 2017; Sawyer, Rodgers & Hart, 2016). A recent study by Denneboom, Bar-Massada & Shwartz (2021), fails to consider the unique species-specific response to overpass width and wrongfully concludes that ungulates prefer narrow overpasses. Our study, along with that of others (e.g., Clevenger & Waltho, 2005) have demonstrated that different species within taxonomic groups often exhibit differing responses to overpass width. Consequently, future studies should investigate the effect of overpass width on specific species or other metrics of overpass success (e.g., biodiversity, biomass).

Scientists and transportation professionals should strive to develop a standardized measure of structure effectiveness to inform future investment in crossing structures (as pointed out by Rytwinski et al. (2015) and van der Grift et al. (2013)). Such investment is likely to increase. For example, in July 2021, the U.S.A. passed the INVEST in America Act, a 5-year highway bill that includes $70 million per year for crossing structures. To guide these future investments, a standardized procedure for measuring crossing effectiveness should be developed that can overcome differing ecological conditions between structures and projects that hamper comparisons. One option that is easily integrated into current monitoring programs with remote cameras is to include remote cameras on wildlife trails a few 100 m away from the structure. With this design, investigators can get a sense of species detection rates between cameras on the structure and those nearby, allowing for a transparent measure of effectiveness that can be compared to hit rates of species nearby. A similar design has been previously suggested by Rytwinski et al. (2015) as part of their plea for increased use of rigorous study designs, such as before-after-control-impact designs, for evaluating road mitigation effectiveness. For example, a structure that has ten elk detections per week and 100 elk detections per week nearby on wildlife trails (i.e., 10:100, or 0.1) would be less effective than a structure that had 80 elk detections per week on the structure and 110 elk detections per week nearby on wildlife trails (i.e., 80:110, or 0.72). The latter structure facilitating movement for more of the locally available elk than the former. When available, incorporating other forms of biological data (e.g., demographic data such as number of adult females with offspring) can help to further determine if the overpass provides demographic and population wide connectivity (Ford, Barrueto & Clevenger, 2017). In Banff National Park, overpass effectiveness in terms of gene flow has also been successfully demonstrated using genetic data for black bears and grizzly bears (Sawaya, Kalinowski & Clevenger, 2014).

Cost effectiveness

The cost of wildlife overpass construction and maintenance is likely the main constraint limiting transportation agency’s ability to meet expert recommendations for overpass dimensions. Various expenses are incurred at all stages of the project including project planning, design selection and construction (McGuire et al., 2021). Cost estimates should also consider maintenance over the entire lifetime of the overpass (i.e., 70 years) as well as end of life expenses. With numerous overpasses built around the globe, transport professionals have identified various innovative ways in which the cost of overpass projects can be reduced while maintaining or even improving ecological effectiveness. For example, buried bridge overpasses save costs when compared to traditional free-standing overpasses while also maximizing soil depth which is favorable for the success of native plant species (McGuire et al., 2021).

In the Netherlands, Sijtsma et al. (2020) found that overpasses present a less cost-effective solution than underpasses. They argue that the high construction costs of overpasses outweigh their absolute benefit to local biodiversity, making underpasses a more cost-effective solution. The authors, however, concede that their threat-weighted ecological quality cost-effectiveness analysis does not differentiate between “different species and nature types”. As a result, in places with more diverse and abundant assemblages of large mammals or with width-sensitive species, overpasses may become more cost effective. Indeed, Ford, Barrueto & Clevenger (2017) performed a cost-effectiveness analysis of grizzly bear crossings at five crossing structure types in Banff National Park, Canada. Using a demographic-specific cost-effectiveness economic model, they found that especially amongst family units, overpasses are more cost effective than underpasses. Amongst singleton bears, the cost effectiveness of overpasses and underpasses were similar. The costs for underpasses used in Ford, Barrueto & Clevenger (2017) included the assumption that the highway was under construction at the time, thus representing a lower cost than if the underpass was added to an existing highway, as is the case for many places mitigation is being considered. The findings between the two studies illustrate how the results of cost-effectiveness analysis may vary at different scales of study, target species assemblage, and stage of highway construction.

A wildlife crossing project being developed along Highway 3 in the southern Rocky Mountains of Canada provides a case-study where transportation professionals, scientists, conservation organizations, industry partners, and First Nations are working to create a cost-effective design in a working landscape. The “Reconnecting the Rockies: BC” project is focused along a 27 km stretch of Highway 3 in southeastern BC, where movement corridors for deer, elk, sheep, moose, bear, wolf, wolverine, and cougar all intersect with a busy highway. The project is planned to feature two purpose-built underpasses, six retrofitted bridges to allow wildlife passage, and an overpass located in the critical Alexander-Michel corridor will be a main feature of the project. The overpass will be the most expensive aspect of the project and multiple designs have been proposed and analyzed for cost. The overpass will span a length of 75 m across two lanes of road and a railway line. The preferred overpass design, which was costed in 2020 for three different widths, 40, 50, and 70 m wide, would respectively cost an estimated $6.2, $7.3, and $9.7 million. Although there is some efficiency of scale as widths increase—with the cost per meter of width decreasing from $0.155 million per meter for the 40 m structure, to $0.139 million per meter for the 70 m structure, the overall increase in price as overpass width increases means that compromises need to be made to accommodate the restricted budget available for this project. While construction has not yet started on this structure, the collaborative group is currently finishing the design phase of the project and favours the 50 m wide structure, which meets the expert recommendations for the target species (a diverse assemblage of large mammals), incorporates some savings with scale, but falls in the middle ground for price option available. The Reconnecting the Rockies example highlights the real-world trade-offs that are required, and the tension between increasing costs and structure effectiveness, which can be optimized through pairing expert design recommendations and innovative engineering support.

Conclusions

Our review finds that transportation agencies are often building structures below expert design recommendations. We suspect the high costs of wide overpasses are a key factor in explaining the high rates of non-compliance of overpasses in Europe and North America. The lack of universal dimensional definitions and poor communication of certain guidelines (i.e., width to length ratios) has likely also contributed to instances of overpass non-compliance. Despite a history of non-compliance to the specific 0.8 width: length guideline in Europe, the width of overpasses around the world should scale accordingly with their length. Importantly, we recommend guidelines also be updated regularly to ensure they accurately reflect the best available science. Indeed, the United States Federal Highway Administration plans to update their Wildlife Crossing Structure Handbook in 2023. More rigorous wildlife overpass effectiveness analysis will require better experimental design (e.g., BACI) and/or additional supporting overpass parameter data such as focusing on population level parameters (e.g., genetic interchange, demographic data on overpass use by breeding females and offspring), adjacent land use and surrounding wildlife densities. We conclude that wide overpasses (~50 m) continue to present important, cost-effective solutions in decreasing the barrier effective of the road (especially when targeting width sensitive species and large assemblages of mammals) but encourage future studies to further explore the specific instances when multiple underpasses and narrower overpasses present more cost-effective solutions.

Supplemental Information

Supplemental Information 1 Overpass inner width in relation to the number of successful wildlife crossings per month.

Dots represent individual overpass structures. Data compiled from transportation agencies and government reports. Species included in analysis: (black bears (Ursus americanus), grizzly bears (Ursus arctos),wolves (Canis lupus), coyote (Canis latrans), cougars (Puma concolor), deer (Odocoileus sp.), elk (Cervus elaphus) moose (Alces alces) crossing rates and the width of 12 overpasses located in western North America. See Supplemental Information Table S7 for details.

Click here for additional data file.

Supplemental Information 2 Generalized linear model coefficients for regressions between overpass inner width and the number of successful wildlife crossings per month for each species.

Error bars are standard errors. Grizzly bear and black bear were significant (p = 0.01497and 0.01706, respectively). Data compiled from transportation agencies and government reports. Species included in analysis: (black bears (Ursus americanus), grizzly bears (Ursus arctos),wolves (Canis lupus), coyote (Canis latrans), cougars (Puma concolor), deer (Odocoileus sp.), elk (Cervus elaphus) moose (Alces alces) crossing rates and the width of 12 overpasses located in western North America. See Supplemental Information Table S7 for details.

Click here for additional data file.

Supplemental Information 3 Parameters of the 120 wildlife overpasses included in review. Min and max shown in brackets.

Click here for additional data file.

Supplemental Information 4 North American Overpass Parameters Reported in Literature.

Click here for additional data file.

Supplemental Information 5 North American Parameters Estimated in Google Earth.

Click here for additional data file.

Supplemental Information 6 European Overpass Parameters Reported in Literature.

Click here for additional data file.

Supplemental Information 7 European Parameters Estimated in Google Earth.

Click here for additional data file.

Supplemental Information 8 Assessment of dimensional guideline compliance for various structures with information gathered from both Google Earth and the Literature for structures with supporting information in the literature.

Click here for additional data file.

Supplemental Information 9 Supporting information for 12 overpass structures included in overpass effectiveness analysis.

Species included in analysis: (black bears (Ursus americanus), grizzly bears (Ursus arctos),wolves (Canis lupus), coyote (Canis latrans), cougars (Puma concolor), deer (Odocoileus sp.), elk (Cervus elaphus), moose (Alces alces) and bighorn sheep (Ovis canadensis) crossing rates and the width of 12 overpasses located in western North America.

Click here for additional data file.

Supplemental Information 10 Test measurements from professional or collegiate football stadiums across the continental U.S.A. to determine relative error of measurements made using the ruler tool in Google Earth Pro 7.3.4.8573 (64-bit).

Click here for additional data file.

Supplemental Information 11 Relative error of Google Earth Pro 7.3.4.8573 (64-bit) ruler tool.

Click here for additional data file.

We thank T. Clevenger whose comments greatly improved this manuscript. We also thank the British Columbia Ministry of Transportation and Infrastructure, Parks Canada, and the Washington State Department of Transportation whose data was used in our analysis.

Additional Information and Declarations

Competing Interests

Author Contributions

Data Availability

The authors declare that they have no competing interests.

Liam Brennan conceived and designed the experiments, performed the experiments, analyzed the data, prepared figures and/or tables, authored or reviewed drafts of the article, and approved the final draft.

Emily Chow conceived and designed the experiments, performed the experiments, authored or reviewed drafts of the article, and approved the final draft.

Clayton Lamb analyzed the data, prepared figures and/or tables, authored or reviewed drafts of the article, and approved the final draft.

The following information was supplied regarding data availability:

All data and associated code are available at GitHub: https://github.com/ctlamb/Wildlife-Overpass-Dimensions.

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
