# Peer review of "Wildlife overpass structure size, distribution, effectiveness, and adherence to expert design recommendations"

_PeerJ, doi:10.7717/peerj.14371_

## Round 0.1 · original submission · Minor Revisions

As you can see both of my reviewers like your paper. So do I — indeed, I appreciate your correcting what is clearly a mistake on my part on a paper I published on this subject last year.

When you submit your revision, I will almost certainly accept it. At PeerJ that means it goes quickly into production. There will not be an opportunity to make corrections. This is an important paper, especially for those of us who are active in creating wildlife corridors of one kind or another.

·

Basic reporting

The field of road ecology, a contemporary hybrid-transdisciplinary form of applied ecology, is now demonstrably entering the early stages of genuine maturity. Having been largely an observational and evidence-limited field of research, road ecology has always been self-consciously aware of its inferential limitations. This has led to a decidedly critical approach, due to the unavoidable strictures imposed by the logistical realities of engineers and transportation planners. It was a source of considerable frustration that replication, among other problems, is rarely sufficient to enable clear conclusions.

In recent times, however, the extraordinary level of road crossing structures being erected all over the world has allowed some of these limitations to be finally be addressed. This study is a rare but extremely welcome example of what can now be undertaken by taking advantage of the number of structures now available globally and the ability to ‘visit’ and measure many aspects of them virtually. This is a timely, broad in scope, data-rich and critically important project with considerable implications for the way that these expensive structures and viewed and how decisions on future implementation can be improved.

Experimental design

The study located 120 wildlife overpasses from around the world although – inevitably – the majority of these were from North America and Europe. These were visited virtually and a series of key parameters were measured. This phase of the study was primarily designed to ascertain the dimensions and to compare these measures to standards recommended by experts and transportation authorities.

For a smaller sample of overpasses, ‘effectiveness’ was ascertained using data on crossing rates by a series of key large mammal species and the relationship between the cost of the structure and the rates of crossings was discerned, as a way of assessing the cost-effectiveness of the structure.

The most severe limitation of the work, however, was the almost non-existent use of statistical verification. It seems that the only analyses were ‘regressions’ though even the form of this was given no details. This is greatly surprising as the data would seem to have been sufficient to allow far more robust assessments to be undertaken.

Validity of the findings

The study found that the majority of overpasses did not meet the minimum criteria of scale recommended by the relevant authorities but concluded that these structures were cost-effective in terms of the primary goal of overcoming the road barrier effect.

These are powerful and critically important findings and should be given considerable prominence. They have immediate implications for the design of wildlife-friendly road designs and will surely become a standard reference of application in the ever-expanding road ecology world.
However, the simplistic and indeed, minimalist approach to statistical assessment must surely undermine these conclusions.

One somewhat technical criticism might be that the parameter employed here for ascertaining the ‘effectiveness’ of a crossing structure was the somewhat simplistic measure of crossing rates by a series of large mammals. The question of how best to evaluate the actual effectiveness of crossing structures is a live issue at present and the measure used here is likely to be regarded as far too broad and biased. In particular, the lack of an appreciation of the genetic effects of the road barrier and whether the reconnection of severed populations can restore previous levels of gene flow is a serious limitation. Nonetheless, as long as the simple parameter employed here is not regarded as more than a useful proxy for reality, its validity can be maintained.

Additional comments

The acquisition of the data from such a large number of structures from around the world is impressive and valuable. This study will undoubtedly become a benchmark for further more detailed similar studies.

·

Basic reporting

The ms is well prepared, clearly written with specific objectives and focus. The pursuit of these objectives are of value to the conservation science community and transportation infrastructure practitioners, particularly in NA.
References and literature pertinent to the objectives were comprehensive. Background context and justification for this work is provided and the rationale explained.

I did not encounter any significant weaknesses or omissions in the ms.

Experimental design

The design is based on the specific objetives, primarily to review the published and non peer review literature to obtain information on dimensions of wildlife overpasses and their efficacy (NA data).
The research questions are clearly explained and justified. The study design is adequate, the methods well described and implications of the results are described in the context of transportation agency practice. I found the research to be accurate and contextual.

Validity of the findings

The results are of high importance to transportation practice in NA and elsewhere. The authors have provided in Conclusions the main findings and current state of practice and highlighted what changes are needed to continue to mitigate impacts of roads on wildlife populations given a range of conservation needs and taxonomic breadth.

I didn't find any areas of ms where authors did not meet high standards for publication.

Additional comments

I enjoyed reading the thorough review of wildlife overpass science. My only comments, albeit minor, are the following:

- L240. it would be interesting to have included >80m in analysis since these structures are not exclusive of animal species needs for large land bridge design. They would have increased sample only slightly (indicated in ms) but I would not have excluded this category as it adds another dimensional level (upper end) to the analysis. The exclusion of mixed use (human/wildlife) overpasses is understandable given the impacts of human use on wildlife use and type of wildlife typically using mixed use OP.

- L422: we should also consider the metric of 'biomass" of wildlife using overpasses in addition to species diversity. Both are indicators of ecosystem integrity and function.

- L444: Adult females with offspring is an additional indicator or type of biological data that can provide important demographic information on OP efficacy. Breeding females with young are the most important metric of success (see Ford et al. 2017), further this method is low cost compared to more intensive and costly methods of genetic analysis of non-invasive sampling.

- L450: the cost of OP is critically important and many times decisive by agencies in design and build. We should be thinking about not only the construction cost but that cost in the context of the amortization period of the OP, i.e., life span of 70-80 years.

- L495-6: Cost is primary focus in much of the ms. Although authors mention innovative engineering support, this subject has not been addressed at all in the ms. There are examples where engineers have purposefully designed novel and innovative OP structures with aim of lowering standard costs. This is the other side of OP design and construction that has to be adopted to practice by bridge engineers in transp agencies. See technical report by McGuire et al. in US Forest Service Gen Tech Report PSW-GTR-267 on innovation strategies in overpass design.

- L498 SLOSS: This debate is nonsensical as the decision of SL or SS is entirely dependent of the target species, specific objectives, landscape and ecosystem. It is entirely context specific and species/taxon driven. There can be no general rule regarding SL or SS, thus discussion and debate is a waste of time. For a specific landscape or region the SL or SS debate could be used to entertain the pros and cons of different planning scenarios. - This section should be reduced considerably or deleted as it did not appear in the objectives but became a side topic while preparing the Disc.

- L531: The FHWA WC handbook is scheduled to be revised in 2023.

- L535: "supporting overpass parameter data" such as focusing on population level parameters including genetic interchange and demographic data on OP use by breeding females and offspring,

---

## Round 0.2 · accepted · Accept

Thanks for making your corrections quickly.